# Acetylcholine-Binding Protein Affinity Profiling of Neurotoxins in Snake Venoms with Parallel Toxin Identification

**DOI:** 10.3390/ijms242316769

**Published:** 2023-11-26

**Authors:** Giulia Palermo, Wietse M. Schouten, Luis Lago Alonso, Chris Ulens, Jeroen Kool, Julien Slagboom

**Affiliations:** 1Centre for Analytical Sciences Amsterdam (CASA), 1012 WX Amsterdam, The Netherlands; giulia.palermo1997@gmail.com (G.P.); w.schouten@nki.nl (W.M.S.); l.lago.alonso@vu.nl (L.L.A.); 2Amsterdam Institute of Molecular and Life Sciences, Division of BioAnalytical Chemistry, Department of Chemistry and Pharmaceutical Sciences, Faculty of Science, Vrije Universiteit Amsterdam, 1081 HV Amsterdam, The Netherlands; 3Laboratory of Structural Neurobiology, Department of Cellular and Molecular Medicine, Faculty of Medicine, KU Leuven, 3000 Leuven, Belgium; chris.ulens@kuleuven.be

**Keywords:** snake venom, neurotoxicity, AChBP bioassay, three-finger toxins, nanofractionation, elapid venom profiling, mass spectrometry

## Abstract

Snakebite is considered a concerning issue and a neglected tropical disease. Three-finger toxins (3FTxs) in snake venoms primarily cause neurotoxic effects since they have high affinity for nicotinic acetylcholine receptors (nAChRs). Their small molecular size makes 3FTxs weakly immunogenic and therefore not appropriately targeted by current antivenoms. This study aims at presenting and applying an analytical method for investigating the therapeutic potential of the acetylcholine-binding protein (AChBP), an efficient nAChR mimic that can capture 3FTxs, for alternative treatment of elapid snakebites. In this analytical methodology, snake venom toxins were separated and characterised using high-performance liquid chromatography coupled with mass spectrometry (HPLC-MS) and high-throughput venomics. By subsequent nanofractionation analytics, binding profiling of toxins to the AChBP was achieved with a post-column plate reader-based fluorescence-enhancement ligand displacement bioassay. The integrated method was established and applied to profiling venoms of six elapid snakes (*Naja mossambica*, *Ophiophagus hannah*, *Dendroaspis polylepis*, *Naja kaouthia*, *Naja haje and Bungarus multicinctus*). The methodology demonstrated that the AChBP is able to effectively bind long-chain 3FTxs with relatively high affinity, but has low or no binding affinity towards short-chain 3FTxs, and as such provides an efficient analytical platform to investigate binding affinity of 3FTxs to the AChBP and mutants thereof and to rapidly identify bound toxins.

## 1. Introduction

Snakebite envenoming is a neglected tropical disease that represents a public health issue mainly in tropical and subtropical regions of the world, with a total number of snakebites per year estimated to be around 5.4 million, resulting in more than 100,000 deaths [1,2,3]. The main pathological effects of snake venoms, which are mixtures of multiple (ca. 50–200) proteinaceous components or toxins, include neurotoxicity, haemotoxicity and myotoxicity [4,5,6]. Currently, the only specific and available therapy for snakebite envenoming is antivenom, which consists of polyclonal antibodies derived from serum or plasma of immunised animals [7]. Antivenoms are specific to the venom(s) used in their manufacture and have limited para-specific efficacy as a result of great venom variation [8]. Furthermore, in addition to their limited para-specific efficacy, current IgG antivenoms are also ineffective in appropriately targeting and neutralising the 3FTxs.

Because of the small molecular size of 3FTxs, they have on average a low immunogenicity and therefore yield IgG antivenoms with often limited effectiveness in neutralisation of the 3FTxs in venoms [9,10]. The 3FTxs are mainly responsible for neurotoxic effects and have high binding affinity for the postsynaptic neuromuscular and neuronal nAChRs, thereby inhibiting the binding of acetylcholine (ACh) and leading to flaccid paralysis. Mostly found in Elapidae snake venoms, 3FTxs are small (~6–9 kDa) disulphide-rich proteins characterised by their three distinct β-stranded loops (or fingers) and consist of two main classes: short-chain 3FTxs (60–62 amino acid residues and four disulphide bonds) and long-chain 3FTxs (66–74 amino acid residues and five disulphide bonds) [11,12]. With 3FTxs being responsible for severe neurotoxic effects and being one of the dominant protein families in elapid venoms [13], the major therapeutic challenge linked to their neutralisation creates space for research on potential therapeutic alternatives other than antivenom.

Along this line, AChBPs have been identified as promising therapeutic targets that are potentially able to neutralise low-molecular-mass neurotoxins such as 3FTxs [14]. Previous research has in fact shown their ability to bind and neutralise long-chain 3FTxs and to generate delayed lethality in mice when administered together with other antivenoms [14,15]. The AChBP is a stable and water-soluble structural homologue of the extracellular domain of the nAChRs. It forms pentameric complexes with a ligand-binding pocket similar to that of the α7-type nAChR [16]. Specifically, the AChBP derived from the mollusc *Lymnaea stagnalis* (Ls-AChBP) shares ca. 26% sequence identity with the extracellular domain of the human α7-nAChR [17], and is considered an excellent model to profile neurotoxicity of nAChR antagonists such as 3FTxs [14,15,18,19]. In order to explore the AChBP’s therapeutic potential, it is of crucial importance to further investigate the 3FTxs in different medically relevant snake venoms for their binding properties to the AChBP. For this reason, analytical methods that allow investigating binding capabilities of both short- and long-chain 3FTxs in elapid snake venoms to the AChBP are needed. As of today, research has been done on methods for so-called high-resolution screening (HRS) in combination with AChBP affinity profiling [20,21]. For this, nano-LC systems with online post-column microfluidic biochemical assays have been developed and demonstrated. These analytical methods allow AChBP-binding affinity profiling with parallel acquired accurate mass assessment of binding ligands after chromatographic separation [22,23,24,25]. They, however, do not allow rapid post-column proteomics analysis of eluting ligands in cases of venom toxins for toxin ID assessment. HRS analytics have been adapted to plate reader format in 1536-well plates for small molecules [26]; however, this has never been developed towards venom profiling. Therefore, such a methodology is needed in combination with parallel high-throughput (HT) venomics in order to achieve both assessment of AChBP toxin binding after chromatographic separation and parallel toxin identification. This new analytical platform, which includes nanofractionation analysis of crude snake venoms, direct post-column AChBP affinity profiling, parallel accurate mass assessment and HT venomics, is needed and presented in this study.

In this study, the venoms of *Naja mossambica*, *Dendroaspis polylepis*, *Ophiophagus hannah*, *Naja kaouthia*, *Naja haje* and *Bungarus multicinctus* were investigated. Of these, the first three are presented in this article and the rest in the Appendix A section. With this methodology, separation and characterisation of toxins from crude venoms was achieved by means of high-performance liquid chromatography–QTOF–mass spectrometry (HPLC-QTOF-MS). Via a post-column flow split (1:9 split ratio) prior to MS detection, the smaller flow fraction was directed to the MS and a larger fraction was sent to nanofractionation in 384-well plates following our current standardised workflow for nanofractionation analytics (see Figure 1). Nanofractionated snake venom toxins were then further analysed offline following our recently presented high-throughput (HT) venomics approach for toxin identification purposes [27]. In parallel, nanofractionated snake venom toxins were profiled for their binding affinity to the Ls-AChBP through fluorescence-based tracer ligand displacement analysis. For this post-column plate reader-based bioassay, the tracer ligand (*E*)-3-(3-(4-diethylamino-2-hydroxybenzylidene)-3,4,5,6-tetrahydropyridin-2-yl)pyridine (DAHBA), which shows enhanced fluorescence in the AChBP-binding pocket, was used [15,19,25,28]. All data measured were finally integrated and correlated with one another for the purpose of rapidly identifying toxins that showed binding affinity to the AChBP. A full overview of the analytical workflow is presented in Figure 1.

## 2. Results

### 2.1. Snake Venom Neurotoxin Binding Profiling to the AChBP

The objective of this study was to present and apply an analytical method to characterise the binding properties of 3FTxs from elapid snake venoms to the AChBP. For this purpose, reverse-phase HPLC separation, nanofractionation on a 384-well plate and parallel detection of snake venom toxins in the separated venoms by MS was performed. Next, a fluorescence-based ligand displacement bioassay (see Section 4.7) was carried out on the fractionated toxins (on the well plate) after vacuum freeze-drying the 384-well plate overnight. This bioassay measures changes in relative fluorescence units between control wells (containing the AChBP and DAHBA) and the wells containing active toxins (i.e., toxins that bind to the AChBP). A decrease in fluorescence is indicative of binding competition, and as such, displacement of tracer ligand DAHBA by one or more snake venom toxins, resulting in formation of the toxin AChBP complex. In total, six snake venoms selected from the Elapidae family were explored in this study to determine binding affinity of both short- and long-chain neurotoxins in these venoms. The venoms included in this study comprised venoms from *Naja kaouthia*, *Naja mossambica*, *Naja haje*, *Ophiophagus hannah*, *Bungarus multicinctus* and *Dendroaspis polylepis*. In this paper, only the data from *Ophiophagus hannah*, *Dendroaspis polylepis* and *Naja mossambica* will be shown to prevent it from containing too much data and keeping it concise. For the interested reader, the results obtained from the remaining snake venoms can be found in the Appendix A section. The snake venoms included in this study were selected based on medical relevance to snakebite and/or taking into account their different abundance of 3FTxs based on the literature [13,29]. Snake venoms of *Naja mossambica*, *Ophiophagus hannah* and *Dendroaspis polylepis* account for high percentages of 3FTxs and present differences in abundance of long- and short-chain 3FTxs. Based on the available literature, the venom from the snake *Ophiophagus hannah* shows high abundance of long-chain neurotoxins and fewer short-chain neurotoxins [30,31,32,33,34], thus predicting results showing high overall activity of the venom. Venom from the snake species *Dendroaspis polylepis* shows presence of several toxins belonging to the long-chain subfamily and other 3FTxs belonging to the short-chain subfamily [35,36,37], thus predicting medium overall activity of the snake venom to be found. Neurotoxicity from the venom of *Naja mossambica* is known to be only derived from short-chain neurotoxins [38,39,40]. As the AChBP is known for binding long-chain 3FTxs and not short-chain neurotoxins, an absence of major bioactivity peaks due to the weak ability of short-chain neurotoxins to bind to the AChBP is expected. The low ability of binding of short-chain 3FTxs to the AChBP has previously been demonstrated by Albulescu et al. [14]. Figure 2 gives an overview of the chromatographic bioassay results obtained from screening the venoms of *Ophiophagus hannah, Dendroaspis polylepis* and *Naja mossambica* through the post-column fluorescence enhancement AChBP bioassay. The chromatographic bioassay results obtained for the venoms of *Naja kaouthia, Naja haje* and *Bungarus multicinctus* are presented in the Appendix A, respectively.

The chromatographic bioassay results of the different snake venoms analysed were then compared qualitatively based on the different activities (i.e., negative peaks) observed. The activities of the different snake venoms were qualitatively evaluated based on the intensity and extent of the negative dip of the observed peaks. The findings from the different snake venoms tested were compared, and as expected, they showed different chromatographic bioassay profiles depending on the eluted bioactive toxins on the well plates after the chromatographic separations. The most potent chromatographic bioactivity profile was recorded for *Ophiophagus hannah* venom, which showed major activity peaks around 10, 17 and 30 min, suggesting the presence of different neurotoxins with high binding affinity to the AChBP. This was to be expected, based on the known high abundance of long-chain 3FTxs in *Ophiophagus hannah* venom from the literature, and is further discussed in Section 2.2 [33]. Bioassay results from *Dendroaspis polylepis* instead showed the presence of a sharp negative split peak in the first 10 min corresponding to high binding affinity of one (or more closely co-eluting) neurotoxin(s) to the AChBP. This major bioactivity peak could be linked to the presence of different long-chain 3FTxs, of which the toxin α-EPTX-Dpp2d has already been proven to have binding affinity towards the AChBP [36]; this is further discussed in Section 2.2. Bioassay results from *Naja mossambica* showed only a few very minor bioactivity peaks along the entire bioassay, with minor peaks appearing at around 25, 32 and 47 min, likely corresponding to snake venom toxins that have low binding affinity to the AChBP. This is to be expected, based on the absence of long-chain 3FTxs in the snake venom of *Naja mossambica* and presence of short-chain neurotoxins and phospholipases A2 (PLA_2_s), which have both previously been shown to have no or low binding affinity to the nAChR mimic AChBP [14,15]. The three elapid snake venoms for which chromatographic bioactivity profiles have now been discussed in this section were next further analysed by means of HT venomics.

### 2.2. High-Throughput (HT) Venomics: Proteomics Identification of AChBP-Bound Toxins

A high-throughput bottom-up proteomics approach, i.e., HT venomics [27], was used for the purpose of identifying those snake venom neurotoxins showing binding affinity to the AChBP after the initial bioactivity profiling through the tracer ligand displacement bioassay. As mentioned in the Materials and Methods section, crude venom samples (from *Ophiophagus hannah*, *Dendroaspis polylepis* and *Naja mossambica*) were fractionated in 384-well plates upon HPLC separation and subsequent mass spectrometric detection. The obtained well plates were subjected to tryptic digestion and analysed by means of nano-LC-MS/MS. From the analyses of these tryptic digests, Mascot search results were obtained that comprised toxin ID and were processed further into protein score chromatograms (PSCs). A detailed description of HT venomics and PSCs is given by Slagboom et al. in their HT venomics study [27]. In short, the protein score retrieved from Mascot for each toxin in each well is plotted on the *y*-axis versus the fractionation retention time of each toxin in each well on the *x*-axis. The data processing from the Mascot data is all automatically done by scripts developed in-house. After the data processing steps using these scripts, protein IDs and their corresponding protein scores are used to build the PSCs, which indicate which proteins are present at a specific retention time and consequently in which fractions. Figure 3 shows a typical example of the PSCs obtained for the venom of *Ophiophagus hannah*. The PSCs in Figure 3 indicate the different proteins retrieved within the venom in different colours, indicating corresponding retention times (*x*-axis) and protein scores (*y*-axis).

Below, Figure 4 shows an overview of the HT venomics results for the three snake venoms investigated by this proteomics approach, focusing on the different abundance of short- and long-chain 3FTxs. For identification of the bioactive toxins, correlation of these data is done by superimposing them with both the chromatographic bioactivity results from the ligand displacement AChBP bioassay and the deconvoluted monoisotopic masses of extracted ion chromatograms (XICs).

For *Ophiophagus hannah* snake venom, a considerable amount of 3FTxs were detected by the HT venomics analysis, which accounted for almost 70% of all proteins detected based on protein scores. The relative toxin abundance in this study is thus based on protein scores. This is certainly not giving actual relative toxin abundance in a venom and can only be used to make comparisons between venoms. Some toxins can yield on average relatively higher protein scores at similar concentrations than other toxins. Of the 3FTx portion in *Ophiophagus hannah* venom, ca. 35% consisted of long-chain 3FTxs and the remainder short-chain neurotoxins. Similarly to *Ophiophagus hannah*, proteomic analysis of *Dendroaspis polylepis* snake venom yielded almost 70% of 3FTxs of the total protein content, this time with a higher abundance (ca. 43%) of short-chain 3FTxs than long-chain ones (ca. 25%). For *Naja mossambica*, results instead showed the presence of ca. 60% 3FTxs, of which all were short-chain. For all snake venoms analysed with HT venomics, proteomics database searches were done in the UniProt database. For the venom sample of *Naja mossambica*, the searches were also done using an available venom gland transcriptomic (species-specific) database. In fact, the transcriptome database allows for searching the venom gland transcriptome of exactly the same species as the species from which the venom proteome is analysed, leading to the most accurate protein matches and usually more toxin identifications compared to searches done in the UniProt database [27,41]. However, these transcriptomics database searches only provide uninformative protein identifier numbers and lack the biological information provided by the UniProt database. For this reason, both the UniProt and the transcriptomics databases were used in parallel in order to obtain complementary information (when a species-specific venom gland transcriptome database was available). The first step towards bioactive toxin identification was correlating the Mascot identities from the protein score chromatograms (PSCs) with bioactivity peaks from the ligand displacement AChBP bioassay. This was achieved by correlating bioactive peaks with PSCs based on matching retention time and peak shape. Considering protein scores, functional activities and subfamilies of the different toxins, it was possible to narrow down the number of toxins corresponding to a bioactivity peak in the AChBP bioassay to a few or even one. In general, taking into consideration the number of short- and long-chain 3FTxs derived from the proteomics results, there was consistency with the different bioactivities observed in the AChBP bioassay, which showed overall low, medium and high binding affinity profiles to the AChBP for *Ophiophagus hannah*, *Naja mossambica* and *Dendroaspis polylepis* respectively. In some cases, it was also possible to exactly match the accurate mass of a toxin to its bioactivity peak by peak shape and retention time correlation of the bioactivity peak and a toxin’s XIC extracted from the HPLC-MS data, but this was obviously not always achieved due to possible mutations, venom variability or other factors. This correlation approach had already proven successful for identification of small proteins (up to 15 kDa) [42]. When an accurate mass of a toxin is known next to its amino acid sequence deduced from the HT venomics results, this information can be used to investigate the toxin further for possible post-translational modifications. In Figure 4, an overview of the superimposed correlated chromatographic data and interpretation with bioactive toxin identification as a focus is shown for the venoms of *Ophiophagus hannah, Naja mossambica*, and *Dendroaspis polylepis*.

For *Ophiophagus hannah* snake venom, different peaks corresponding to activity were observed. Through data correlation, it was possible to match bioactivity peaks to different protein IDs in the protein score chromatograms, in order to identify the bioactive toxins, as shown in Figure 4A. Furthermore, it was also possible to assign two accurate masses to the bioactivity peaks at approximately 10 and 23 min, indicating the presence of snake venom neurotoxins binding to the AChBP. The major activity peak eluting at 10 min was correlated with the protein with an *m*/*z* value of 1404.6185^4+^ for the monoisotopic peak and a deconvoluted mass of 7903.57 Da, corresponding to the Mascot hit and UniProt entry Q53B58—Alpha-elapitoxin-Oh3a retrieved from HT venomics. The second minor activity peak at approximately 23 min was correlated with a toxin with an *m*/*z* value of 1881.8476^4+^ for the monoisotopic peak and corresponding to an accurate mass of 7518.34 Da, which coincides with the Mascot identity Alpha-elapitoxin-Oh2b (UniProt entry P82662) deduced from the HT venomics results. In the PSC of the Mascot identity Alpha-elapitoxin-Oh2b, two peaks with similar protein scores were detected at both ca. 23 and ca. 30 min. However, it was possible to identify the peak at 30 min as a “false match”, likely corresponding to a toxin with a similar amino acid sequence, and spot the other peak at 23 min as the correct match, as an XIC was detected in the MS chromatograms that matched the bioactivity and the accurate mass of the toxin, making the identification unambiguous. XICs of the aforementioned *m*/*z* values corresponding to bioactivity peaks and from there to neurotoxins with binding affinity to the AChBP are shown in Figure 4(A-iii). This binding affinity of both Alpha-elapitoxin-Oh3a and Alpha-elapitoxin-Oh2b to the AChBP indicates the role of these toxins as potent binders to nAChRs, as previously reported [34,43]. Both toxins belong to the 3FTxs family, long-chain subfamily and type II alpha neurotoxin sub-subfamily. Binding of these long-chain neurotoxins to the AChBP is in fact believed to be linked to the presence of a fifth disulphide bond in the second loop of the protein [44,45], which plays a key role in allowing binding to nAChRs and the AChBP. With Alpha-elapitoxin-Oh3a and Alpha-elapitoxin-Oh2b having a central role in neuromuscular paralysis and in general neurotoxic effects generated after envenomation from *Ophiophagus hannah* [43], their binding to the AChBP shows the potential to use the nAChR mimic AChBP as a new snakebite treatment candidate for the neutralisation of long-chain 3FTxs, as previously reported [14]. For the smaller activity peak appearing at ca. 16 min and the minor activity peak at ca. 20 min, it was not possible to assign exact toxin IDs due to the presence of many toxins with PSC peaks of closely co-eluting toxins found in that timeframe with HT venomics, which all had similar protein scores. As Figure 4(A-v) shows, most of these co-eluting toxins, e.g., 3L22H (long neurotoxin LNTX-2 homologue), 3L22 (long neurotoxin 2), and 3L237 (long neurotoxin OH-37), belong to the 3FTx class, long-chain subfamily, and type II alpha neurotoxin sub-subfamily. Also, for the activity peak appearing at ca. 30 min, it was not possible to assign one definitive toxin ID. However, the peak corresponding to the “fake match” of 3L26 (Alpha-elapitoxin-Oh2b) is eluting in this timeframe, and this could probably be linked to the presence of a toxin with sequence similarities to 3L26 that could therefore be responsible for this activity peak at 30 min. Furthermore, of the different Mascot identities of toxins eluting in that timeframe, many corresponded to long-chain alpha neurotoxins, all probably having affinity to the AChBP. Therefore, it is also a possibility that the bioactivity peak was a contribution of several co-eluting toxins. This again confirms the ability of toxins belonging to the long-chain 3FTx family of binding to the AChBP, as documented previously [14].

For the venom derived from *Dendroaspis polylepis*, only one major activity peak was observed in the first 10 min of the bioassay, at around 9.3 min, as described in Section 2.1. For this reason, to not complicate the PSCs in Figure 4(B-v) too much, only the first 20 min of the proteomics results are shown; full proteomics results can be found in the Appendix A. In this case, the major activity peak (corresponding to high binding affinity to the AChBP), was correlated with proteomics and mass spectrometric data, as shown in Figure 4B. One main Mascot identity (DENPO_3L24) was found at the same timeframe of the peak indicating activity, with similar peak shape, which corresponded to Alpha-elapitoxin-Dpp2d (UniProt entry C0HJD7), a long-chain three-finger toxin with 72 amino acids belonging to the type II alpha neurotoxin family. Both this toxin and its dimer have already been documented as potent inhibitors of human α7-nAChR and high-affinity binders to the AChBP in previous research from Anderson Wang et al. [36]. Whilst proteomics analysis suggests the presence of Alpha-elapitoxin-Dpp2d eluting in the timeframe where activity is spotted in the ligand displacement AChBP bioassay, it was not possible to match the exact toxin mass to the accurate mass from the MS data. This was seemingly due to post-translational modifications of the toxin and intraspecific venom variation. However, as the AChBP bioassay from the venom of *Dendroaspis polylepis* in Figure 4B shows, after approximately 10.4 min, proteomics results indicate the presence of mainly 3FTxs belonging to the short-chain subfamily, and this agrees with the low ability/inability of this class of toxins to bind to the AChBP, as previously reported [14].

For *Naja mossambica* snake venom, minor activity peaks related to low binding affinity to the AChBP were observed, as shown in Figure 4(C-iv). For the first activity peak appearing at ca. 22 min, correlated proteomic data indicate the presence of short-chain 3FTxs. Among those, we detected 3SA5 (cytotoxin 5), 3SA2 (cytotoxin 2) and 3SA4 (cytotoxin 4), all belonging to the short-chain subfamily and type IA cytotoxin sub-subfamily. However, due to co-elution of these peaks in the same timeframe, it was not possible to exclusively identify a toxin responsible for the bioactivity. It is most likely that this minor bioactivity peak corresponding to low-binding affinity to the AChBP is correlated with the typical low binding affinity of short-chain 3FTxs to the AChBP, as previous research shows [14]. For the second minor bioactivity peak at ca. 32 min, the HT venomics analysis indicates the presence of PLA_2_ toxins in the timeframe corresponding to the minor bioactivity, suggesting low binding affinity of these toxins to the AChBP. In that timeframe, the toxins PA2B3 (basic phospholipase A2 CM-III), PA2B2 (basic phospholipase A2 CM-II), and PA2A1 (acidic phospholipase A2 CM-I) were detected in the proteomics results and two 3FTxs detected when using the transcriptomics database (full data available under Appendix A). Even though this did not make it possible to distinguish which of the two classes of toxins was responsible for activity, it suggests that a toxin belonging to either the 3FTx or phospholipase A_2_ toxin family could be responsible for minor activity and therefore low binding affinity to the AChBP, as already documented in previous research for these toxin families [14,15]. For the last minor bioactivity peak that eluted at approximately 46 min, proteomics analysis indicates the presence of proteins belonging to the 3FTxs class, short-chain subfamily and type IA cytotoxin sub-subfamily in that timeframe. It was possible to link this bioactivity peak to a toxin with an *m*/*z* value 1705.5877^4+^ for the monoisotopic peak and a corresponding mass of 6814.30 Da. The XIC of 1705.5877^4+^ is shown in Figure (4C-iii). The toxin corresponding to the minor bioactivity peak was identified as “Cytotoxin 1” (UniProt entry P01467), and its presence in that timeframe was confirmed, matching the previously mentioned accurate mass (6814.30) to its equal exact mass that was calculated based on the amino acid sequence. Finding binding affinity of cytotoxin 1 to the AChBP at first seems surprising taking into consideration the differences in functionality and protein sequence compared to α-neurotoxins, typically responsible for higher neurotoxicity and characterised by high binding affinity to the AChBP [46]. However, this short-chain (60 amino acids) cytotoxin had already shown medium binding activity to the AChBP in previous research by Otvos et al. [47], suggesting low neurotoxic activity and thus confirming our findings.

## 3. Discussion

This study aimed at investigating binding affinity of snake venom neurotoxins to the AChBP and to identify these bioactive toxins. The methodology used included separation and nanofractionation of snake venom toxins from crude venom by means of reverse-phase HPLC-MS, the high-throughput bottom-up proteomics approach HT venomics for identification of separated venom toxins, and a fluorescence enhancement-based ligand displacement bioassay for binding profiling of toxins to the AChBP. The current method demonstrated that the AChBP binds long-chain 3FTxs with high affinity and short-chain 3FTxs with no or low affinity, consistent with results from previous studies [14,15,19,44,47], and as expected from known binding specificities of short- and long-chain 3FTxs to the human α7 nAChR [48]. The presence of a fifth disulphide bond in long-chain 3FTxs is in fact believed to have an impact on their biological activity and influence the increased affinity (ca. up to 10^4^ times higher) of these toxins to the homopentameric neuronal α7 receptor compared to short-chain 3FTxs [49,50], which have also shown rapid dissociation from human nAChRs [46]. By making use of the current method and through correlation of protein score chromatograms and an AChBP bioassay, it was possible to identify toxins responsible for bioactivity. Results from three venoms from the Elapidae family show that long-chain alpha neurotoxins such as alpha-elapitoxin-Oh3a and alpha-elapitoxin-Oh2b from the venom of *Ophiophagus hannah* efficiently bound to Ls-AChBP with high affinity, whereas short-chain neurotoxins such as cytotoxin 1 from the venom of *Naja mossambica* have low binding affinity for the AChBP. Furthermore, the highest overall activity was encountered in the bioassay for snake venoms with high abundance of long-chain 3FTxs. No or little bioactivity was detected in chromatographic bioassay timeframes corresponding to elution of short-chain 3FTxs, which currently remains a limitation for AChBP as a target. The results obtained in this study demonstrate the potential of this methodology to combine snake venom neurotoxicity profiling and identification in a high-throughput fashion for with having binding affinity towards the AChBP. Nevertheless, further rapid toxin isolation procedures would represent an important next step for allowing the study of actual toxin binding modes of bioactive toxins to the AChBP, as well as quantification of the observed bioactivities. Furthermore, together with previous research on the different binding modes of AChBP and short-chain 3FTxs [48], this can aid in the development of engineered AChBP mutants that show enhanced binding affinity to short-chain 3FTxs. This will consequently alleviate the limitation inherent to AChBP as a target. Higher binding affinity of these new AChBP mutants for short-chain 3FTxs could open up AChBP as candidate snakebite treatment for neutralising short-chain 3FTx next to the long-chain ones. For the long-chain ones, this has already been demonstrated to decrease lethality generated by neurotoxic snake venoms in mice when administered together with antibody-based antivenom [14]. The employment of the AChBP as a candidate therapeutic agent could therefore assist in helping to overcome challenges linked with current IgG antivenoms in terms of targeting weakly immunogenic 3FTxs [3,10].

## 4. Materials and Methods

### 4.1. Chemical and Biological Reagents

All solvents and chemicals used in this research were of analytical grade. Water was purified with a Milli-Q Plus system (Millipore, Amsterdam, The Netherlands). Acetonitrile (ACN; ULC/MS grade), trifluoroacetic acid (TFA) and formic acid (FA) were purchased from Biosolve (Valkenswaard, The Netherlands). Ammonium bicarbonate, iodoacetamide, β-mercaptoethanol, sodium chloride, potassium phosphate monobasic, sodium phosphate dibasic, Trizma^®^ base and proteomics grade recombinant trypsin were purchased from Sigma-Aldrich (Zwijndrecht, The Netherlands). Pierce™ Protein-Free T20 (TBS) Blocking Buffer was purchased from Thermo Fisher Scientific (Breda, The Netherlands). (±)-Epibatidine was purchased from Tocris Biosciences (Bristol, UK). α-Bungarotoxin from Bungarus multicinctus was purchased from Merck KGaA (Darmstadt, Germany). (E)-3-(3-(4-diethylamino-2-hydroxybenzylidene)-3,4,5,6-tetrahydropyridin-2-yl) (DAHBA) was synthesised in-house at the Medicinal Chemistry Division within the Chemistry and Pharmaceutical Sciences Department of VU Amsterdam. Ls-AChBP (from snail species *Lymnaea stagnalis*) was obtained from the Laboratory of Structural Neurobiology within KU Leuven. In brief, Ls-AChBP was produced in *Sf*9 insect cells using a baculovirus-based expression system. The secreted His-tagged protein was purified from the cell medium with affinity chromatography using Ni Sepharose 6 Fast Flow agarose beads from Cytiva Life Sciences (Hoegaarden, Belgium). Fractions containing Ls-AChBP were identified using a Coomassie-stained SDS-PAGE protein gel. These fractions were then pooled, concentrated and further purified using a Superose 6 Increase 10/300 GL size exclusion column from (Cytiva Life Sciences (Hoegaarden, Belgium) in buffer containing 20 mM TRIS, pH 8.0 and 300 mM NaCl. Fractions corresponding to the pentameric protein were pooled and concentrated to ~6 mg/mL. In the current study, different snake venoms from the Elapidae family were used and analysed. *Naja mossambica* (Mozambique spitting cobra), *Naja haje* (Egyptian cobra) and *Dendroaspis polylepis* (Black mamba) were sourced from LSTM. *Ophiophagus Hannah* (king cobra) and *Naja kaouthia* (monocled cobra) were sourced from the historical collection of F.J. Vonk, and *Bungarus multicinctus* (many-banded krait) was sourced from the historical collection of prof. Kini. Lyophilised snake venoms were stored at −80 °C until reconstitution with Milli-Q water to have final 5 mg/mL stock solutions, which were then aliquoted and stored at −80 °C until use.

### 4.2. HPLC-MS and Nanofractionation of Snake Venom Toxins

All snake venom samples were analysed through liquid chromatographic separation, parallel post-column nanofractionation and mass spectrometry (MS) detection. For each snake venom sample, the entire process was repeated twice. An Agilent 1260 Infinity II LC System by Agilent Technologies Netherlands B.V. (Amstelveen, The Netherlands) was used to perform reverse-phase high-performance liquid chromatography (HPLC). All separation parameters were controlled using Agilent OpenLab CDS version 2.7 software. A volume of 50 μL per snake venom sample was injected using the 1260 Infinity II Multisampler and the flow rate was set at 0.5 mL/min. For venom separation, a 4.6 mm × 100 mm Waters XBridge Peptide BEH C_18_ analytical column with 300 Å pore size and 5 µm particle size was used. The mobile phases comprised eluent A (98% H_2_O, 2% ACN, 0.1% TFA) and eluent B (98% ACN, 2% H_2_O, 0.1% TFA). The gradient used for separation of the elapid venoms consisted of a linear increase in mobile phase B from 1% to 20% in 5 min followed by an increase from 20% to 40% B in 55 min and subsequent increase in mobile phase B from 40% to 90% in 4 min. This was followed by isocratic elution for 5 min at 90% B. Finally, starting conditions (1% B) were reached linearly in 1 min and the column was equilibrated for 10 min at 1% B. After column separation and detection with a 1260 Infinity II Variable Wavelength Detector, the effluent volume was split in two different fractions using an analytical adjustable flow splitter (from Analytical Scientific Instruments US), one consisting of 90% of the volume and the other one consisting of 10% of the volume. The larger fraction (90% of the volume) was sent to a FractioMate^TM^ FRM100 nanofraction collector (VU, Amsterdam, The Netherlands). Using this system with FractioMater Software Version 2.0.0.21, the LC fractions were collected onto black F-bottom 384-well microplates (Greiner Bio One, Alphen aan den Rijn, The Netherlands). Eluent was collected in the wells in a column-by-column fashion, with an up-to-down serpentine spot pattern. A fractionation spotting time of 12 s for each fraction was used. After fractionation, the 384-well plates containing the collected fractions were fully dried overnight (for approximately 16 h) with a Christ Rotational Vacuum Concentrator RVC 2-33 CD plus (Salm en Kipp, Breukelen, The Netherlands) in order to remove all volatile constituents. The fully dried plates were then stored at −20 °C or −80 °C before use for tryptic digestion or the fluorescence enhancement-based AChBP ligand displacement bioassay. The smaller fraction (10% of the volume) was sent to a mass spectrometer for intact toxin analysis. Detailed information on MS parameters is given in Section 4.3.

### 4.3. Mass Spectrometry Analysis of Intact Toxins

Mass spectrometric analysis of all snake venoms was performed using a MaXis QTOF mass spectrometer (Bruker Daltonics, Bremen, Germany) fitted with an electrospray ionization source (ESI) interface in positive ion mode. Instrument parameters were controlled using OtofControl Software Version 5.2 (Bruker, MA, USA). The following optimised parameters were used: gas temperature was 220 °C, capillary voltage 4.5 kV, gas flow 8.0 L/min and nebulizer pressure 1.5 Bar. Spectra were acquired at a rate of 2 Hz in the range of 800 to 5000 *m*/*z*. All spectra were visualised and managed through the Bruker Compass DataAnalysis Software Version 5.2 (Bruker Daltonik GmbH, Bremen, Germany).

### 4.4. In-Solution Tryptic Digestion

HPLC-MS and nanofractionation of each snake venom sample was done in duplicate. Therefore, two fractionated and fully dried 384-well plates were obtained for each venom. One of the two plates was then used for in-solution tryptic digestion and subsequent nanoLC-MS/MS analysis for protein identification. For this purpose, 25 μL of reduction buffer (25 mM ammonium bicarbonate and 0.05% β-mercaptoethanol, pH 8.2) were added to all wells using a ThermoFisher Multidrop pipetting robot (Thermo Fisher Scientific, Ermelo, The Netherlands). After this, the plates were incubated for 15 min at 95 °C in a Hewlett Packard HP 6890 GC System, Agilent Technologies Netherlands B.V. (Amstelveen, The Netherlands). The well plate was then cooled down to room temperature. After this, 10 μL of alkylating agent (12.5 mM iodoacetamide in H_2_O) were added to the wells with the same pipetting robot. Next, the well plate was incubated for one hour at room temperature in the dark. Finally, a stock solution of Trypsin (1 μg/μL in 50 mM Acetic Acid) was diluted 100 times in 25 mM ammonium bicarbonate to reach a final concentration of 0.01 μg/μL; 10 μL of this solution were added to the wells using a repetitive pipette and subsequently the plates were incubated at 37 °C overnight. The day after, the plates were centrifuged at 1000 rpm for 1 min in an Eppendorf Centrifuge 5810 R and finally 10 μL of quenching agent (1.25% formic acid in H_2_O) were added to the wells using the Multidrop pipetting robot. Next, the plates were analysed using nanoLC-MS/MS or stored at −20 °C until analysis.

### 4.5. NanoLC-MS/MS Analysis

Separation of the tryptic digests of all snake venoms was achieved using an UltiMate 3000 RSLCnano system (Thermo Fisher Scientific, Ermelo, The Netherlands). Its UltiMate™ WPS-3000TPL/PL RSLCnano Well Plate Autosampler allowed for direct sampling from the 384-well plates. Volumes of 1 μL per sample were injected and separated using a 150 mm × 75 µm Acclaim™ PepMap™ 100 C_18_ HPLC Column with 100 Å pore size and 2 µm particle size in combination with a 5 mm × 0.3 mm Acclaim™ PepMap™ 100 C_18_ trapping column, with 100 Å pore size and 5 µm particle size, obtained from Thermo Fisher Scientific. The column was stored in the column oven compartment at 45 °C. The mobile phases comprised eluent A (98% H_2_O, 2% ACN, 0.1% FA) and eluent B (98% ACN, 2% H_2_O, 0.1% FA). The gradient used for separation of the digests consisted of isocratic elution for 3 min at 1% B, followed by a linear increase from 1% to 40% B in 7.5 min, subsequent increase from 40% to 85% B in 0.1 min, followed by isocratic elution for 0.7 min at 85% B. Finally, starting conditions (1% B) were reached linearly in 0.2 min and the column was equilibrated for 3.7 min at 1% B. Detection was performed using a MaXis QTOF mass spectrometer (Bruker Daltonics, Bremen, Germany) fitted with a Bruker Captive spray source interface in positive-ion mode. Instrument parameters were controlled using OtofControl software Version 5.2 (Bruker, MA, USA). The following optimised parameters were used: gas temperature was 150 °C, capillary voltage 1.6 kV, gas flow 3.0 L/min and nanoBooster (100% ACN) pressure 0.20 Bar. Spectra were acquired at a rate of 2 Hz in the range of 50 to 3000 *m*/*z*. MS/MS spectra were obtained using collision-induced dissociation (CID) in data-dependent mode using 10 eV collision energy.

### 4.6. NanoLC-MS/MS Data Processing

All data files obtained for each venom after nano-LC-MS/MS analysis of tryptic digests were converted into MGF files using the Bruker Compass Data Analysis Software (version 5.0) and the ProcessWithMethod function; all method parameters can be found in the Appendix A. After this, all resulting MGF files were processed in one batch using Mascot Daemon software (version 2.3.3, Matrix Science, London, UK), which automated data files’ submission to the in-house Mascot server (version 2.3.02) [51]. The searches were done on two different databases: the UniProt database (containing only Serpentes (snakes) accessions) for the venoms of *Naja mossambica, Ophiophagus hannah, Dendroaspis polylepis, Naja kaouthia, Naja haje and Bungarus multicinctus*, and venom gland transcriptomic databases that were species-specific for the venom of *Naja mossambica*. All used Mascot search parameters are described in detail by Slagboom et al. [27]. After running the database searches with the Mascot Daemon software, information on the Mascot search result such as date and specific job number in the search log are gathered in a Mascot Export template (.xlxs format) for each analysed venom. The resulting Mascot Export template files for each venom were then used as input for the first in-house developed script using *R* software for Windows Version 4.2.0. Using this first script, comma-separated values (CSV) files, that contain all information obtained from the database search, were extracted from Mascot search logs. Each separate CSV file corresponds to a single sample (well) in the well plate. Further information on the premade Excel template and first script can be found in the Appendix A. After this, script 2 was used to extract relevant information from the single CSV files obtained from Script 1, which contain information about all toxins found separately in each well, and merge them into one single Excel file for each venom. This Excel file contains information about toxins found in all the wells (together with protein accession, protein score, sequence coverage, protein description, full protein sequence, found peptide sequences and a link to the original Mascot search) for the nano-LC-MS/MS analysis results of each well in a well plate. All these Excel files and the second script can be found in the Appendix A. Next, the third script (found in the Appendix A) was used to plot retention time (*x*-axis) against protein score (*y*-axis) for each individual venom protein found in the wells to generate protein score chromatograms (PSCs). The final script (found in the Appendix A) takes information from Script 3 generated files, merging all x values (retention times) and y values (protein scores) of all venom proteins into one single Excel file. Finally, this file was used to plot retention times vs. protein scores of every venom toxin using GraphPad Prism version 8.0.1 (GraphPad Software, Inc., San Diego, CA, USA). All scripts and files can be found in the Appendix A section and further information on the data processing steps for high-throughput venomics is described in detail by Slagboom et al. [27].

### 4.7. Fluorescence Enhancement-Based AChBP Ligand Displacement Bioassay

The duplicate of the aforementioned fractionated and fully dried 384-well plate (see Section 4.2) for each venom was used for the fluorescence enhancement-based AChBP ligand displacement bioassay. This bioassay allows investigation of the binding affinity of snake venom toxins to the AChBP by displacement of the DAHBA tracer ligand. For this purpose, the high-affinity compounds epibatidine and bungarotoxin were used as positive controls in all evaluation experiments. The plates were used directly after freeze-drying overnight or after letting them reach room temperature if previously stored at −20 °C. Incubation buffer was prepared (1 mM KH_2_PO_4_, 3 mM Na_2_HPO_4_, 0.16 mM NaCl, 20 mM Trizma base at pH 7.5 in MilliQ water with 10% Pierce™ Protein-Free T20 (TBS) blocking buffer) and then used in the bioassay and for dilutions. Solutions of AChBP, DAHBA, bungarotoxin and epibatidine were prepared either in incubation buffer or water. A stock solution of 1 µM AChBP (in water) was diluted in incubation buffer to reach a final concentration of 100 nM. A stock solution of 125 µM bungarotoxin (in water) was diluted in water to reach final concentration of 1 µM. Epibatidine stock solutions of 1 mM were created by dissolving the solid component in DMSO. Two epibatidine solutions were prepared from the stock solution at concentrations of 100 µM and 1 µM, respectively, in incubation buffer. For DAHBA, stock solutions of 1 mM in DMSO were prepared and then diluted to 100 µM and 1 µM in incubation buffer; from this, a dilution in incubation buffer was performed to achieve a final concentration of 300 nM. After this, controls were added in duplicate to the first six wells of column 1 of each 384-well plate. For this, the first two wells were filled with 30 µL of 100 nM AChBP, 30 µL of 300 nM DAHBA and 30 µL of incubation buffer; the following two wells with 30 µL of 100 nM AChBP, 30 µL of 300 nM DAHBA and 30 µL of 1 µM bungarotoxin. The last two wells were filled with 30 µL of 100 nM AChBP, 30 µL of 300 nM DAHBA and 30 µL of 1 µM epibatidine. The ten remaining wells in the first column of the well plate were left empty. This was decided based on there being no compounds eluting in the first circa 3.5 min of the chromatographic separation. Next, from column 2 onwards, the wells of the well plate in which eluent from a fractionation was collected were filled with assay solution. For this, 90 µL portions of a mixed solution of 100 nM AChBP, 300 nM DAHBA and incubation buffer (in 1:1:1 volume ratios) were added to each well using a ThermoFisher Multidrop pipetting robot. The black Greiner 384-well plates were then immediately measured using a Varioskan™ Flash Multimode Reader (Thermo Fisher Scientific, Ermelo, The Netherlands). SkanIt software Version 7.1 (Thermo Fisher Scientific, Germany) was used for instrument control and parameter selection. Measurement mode was chosen to be in serpentine up-to-down fashion, following the same nanofractionation pattern and the temperature was set at 25 °C. An endpoint fluorescence measurement was performed after the plate was incubated with the assay solution for 15 min at RT. Excitation and emission wavelengths of 485 and 520 nm were chosen as optimal conditions for the bioassay based on previous research by Kool et al. [25], and excitation bandwidth of 12 nm was used. All data from the fluorescence measurements were then extracted in the form of Excel files to facilitate plotting via Excel or Graphpad Prism version 8.0.1. The information retrieved from the measurement was relative fluorescence units (RFU) values for each well of the 384-well plate. Using an Excel template, all RFU values corresponding to the wells were assigned to their corresponding fractionation retention times provided by the FractioMate ^TM^ FRM100 system. This was done in order to correlate each snake venom fraction in the well plate to its corresponding RFU value. Measurement (i.e., retention) times (*x*-axis) and RFU values (*y*-axis) were then plotted on a graph to check for binding affinity of venom toxins to the AChBP, with a decrease in RFU values corresponding to displacement of tracer ligand DAHBA and thus binding of a snake venom toxin to the AChBP. The first few minutes of the chromatographic bioassay results are not shown in the figures, as no toxins eluted in this timeframe and the first column was used for controls.

## Figures and Tables

**Figure 1 ijms-24-16769-f001:**
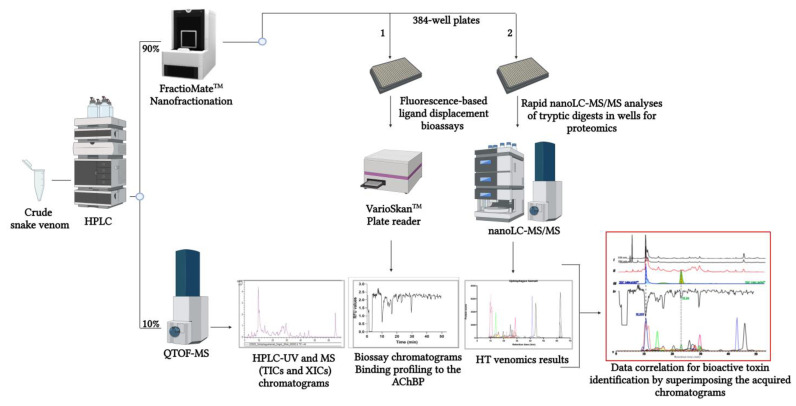
Graphical overview of the analytical workflow followed in the current study. Crude snake venoms follow a standardised nanofractionation analytics workflow, which includes HPLC separation of venom toxins followed by a flow split where 10% of the effluent reaches the mass spectrometer (QTOF-MS) for intact toxin analysis and 90% is sent to a high-resolution nanofractionator that collects HPLC fractions onto 384-well plates. Two replicate well plates with separated venom toxins are generated for each snake venom, and after eluent evaporation using a vacuum centrifuge, they are employed for either one of two different purposes. One well plate is used for tryptic digestion and subsequent bottom-up proteomics analysis for protein identification by HT venomics, while the other well plate is used for the fluorescence-enhancement based ligand displacement bioassay with tracer ligand DAHBA for toxin binding profiling to the AChBP. Results from the different analyses (i.e., HPLC-UV, HPLC-MS, so-called bioassay chromatograms, and HT venomics so-called protein score chromatograms) are correlated by superimposing the chromatographic data with the purpose of identifying snake venom toxins binding to the AChBP.

**Figure 2 ijms-24-16769-f002:**
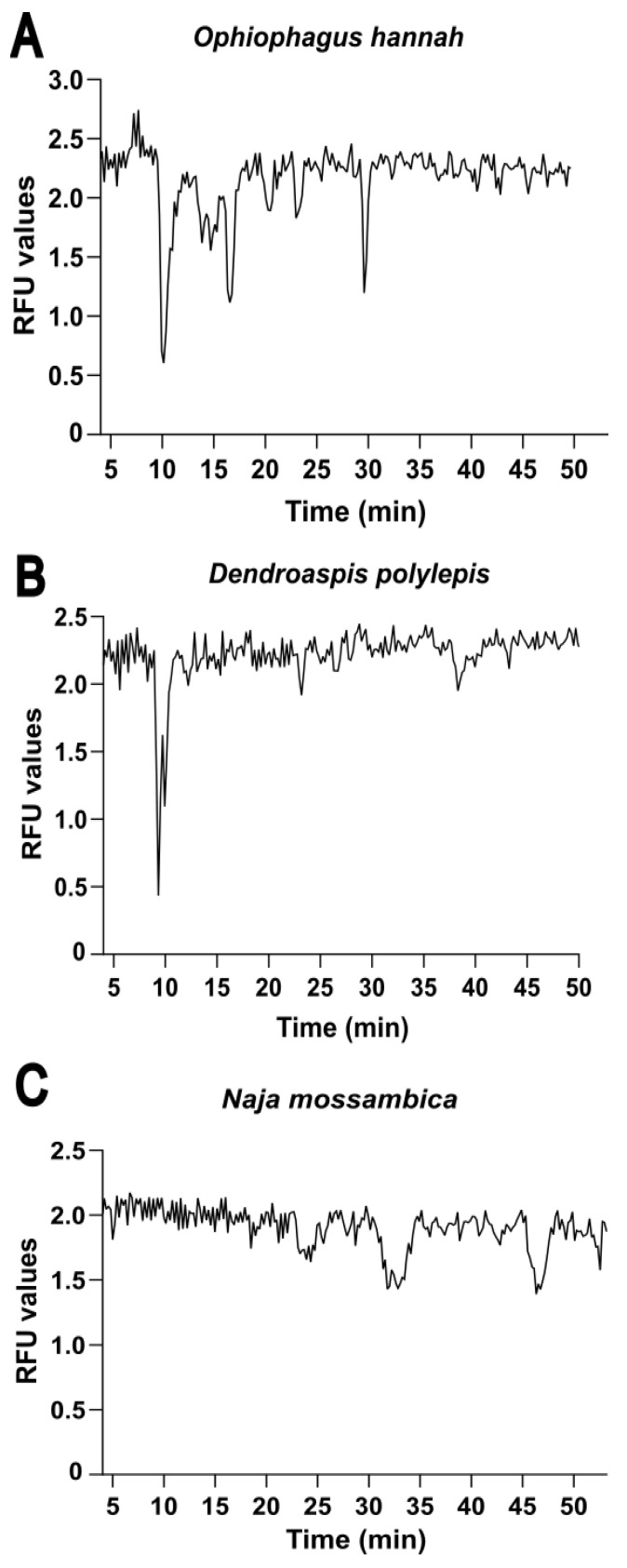
Fluorescence-based ligand displacement bioassay for neurotoxin-binding profiling of the venoms of (**A**) *Ophiophagus hannah*, (**B**) *Dendroaspis polylepis*, and (**C**) *Naja mossambica*. The bioassay allows investigation of the binding affinity of different snake venom toxins to the target AChBP directly after chromatographic separation of the toxins in the venoms under study. For the bioassay chromatograms in the figure, retention time of fractionation is plotted on the *x*-axis versus bioassay readout on the *y*-axis with a connecting line between the measurement points. A decrease in fluorescence is indicative of competition displacement of the tracer ligand DAHBA from the AChBP by eluted toxins.

**Figure 3 ijms-24-16769-f003:**
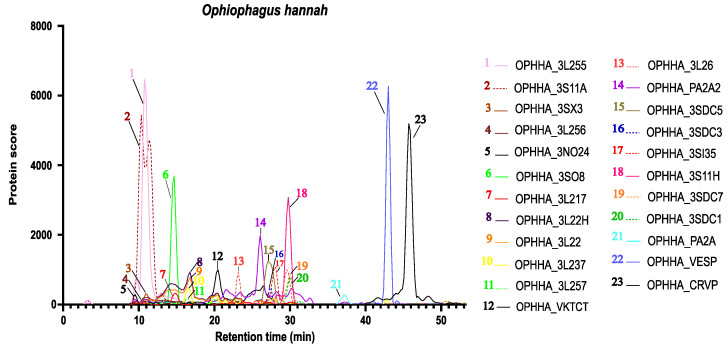
Typical protein score chromatograms (PSCs) generated for identification of the toxins in snake venom. In the figure, only several of all toxins retrieved by HT venomics are plotted so asnot to complicate the figure too much with too many PSCs. The PSCs of all toxins retrieved can be found in the Appendix A. The *x*-axis indicates retention time in minutes and the *y*-axis gives protein scores for each toxin plotted in the figure for each well it was found in by proteomics. Accession numbers (unique protein identifiers) of the toxins for the plotted PSCs are given in the legend on the right panel. Similar results for the venoms of *Naja mossambica*, *Dendroaspis polylepis*, *Naja kaouthia*, *Naja haje*, *and Bungarus multicinctus* are given in the Appendix A.

**Figure 4 ijms-24-16769-f004:**
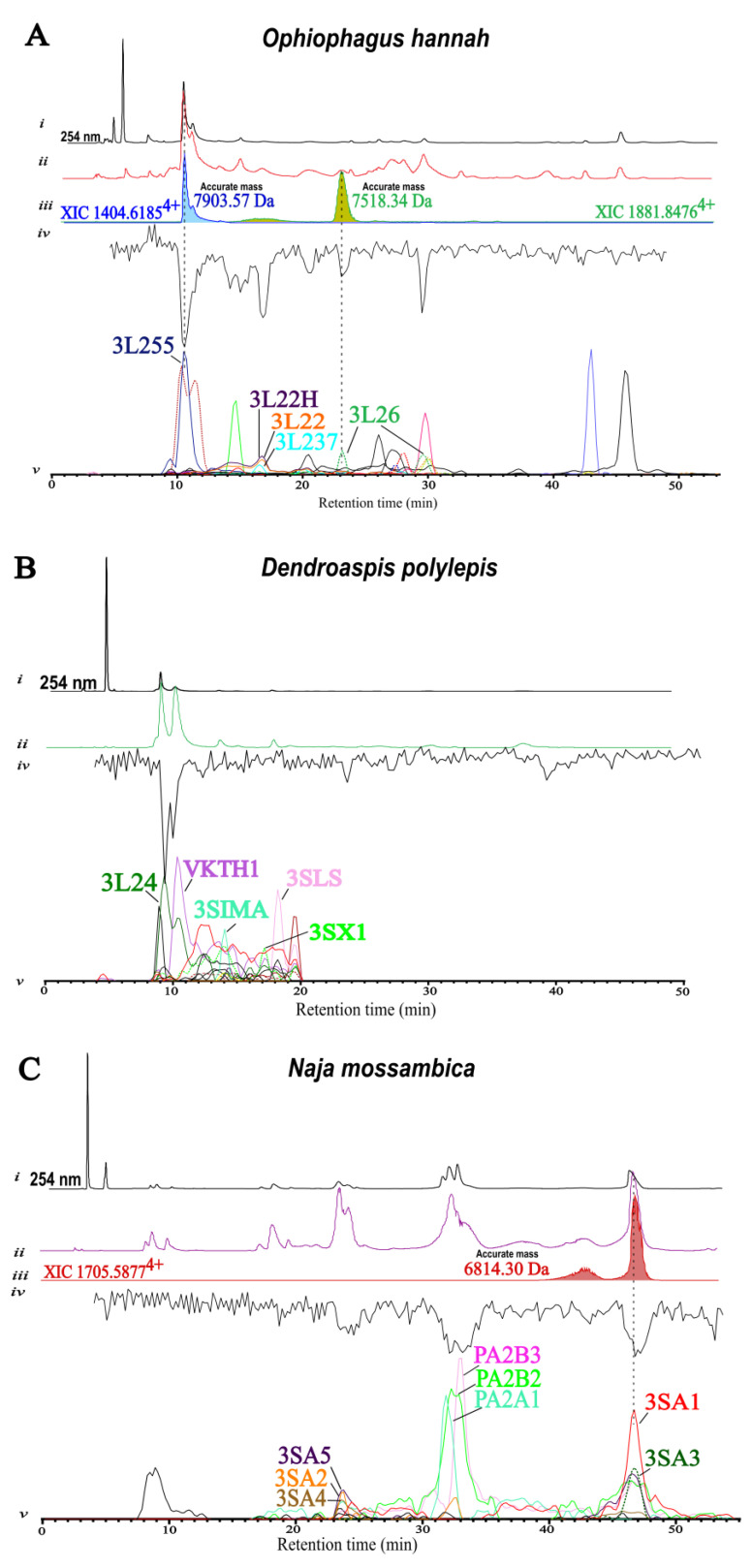
Identification of toxins with binding affinity towards the AChBP by correlating the AChBP bioactivity chromatographic peaks in terms of peak shape and retention time with XIC peaks from the MS data and with PSCs from the HT venomics data for the venoms of (**A**) *Ophiophagus hannah*, (**B**) *Dendroaspis polylepis*, and (**C**) *Naja mossambica*. In the figures, the acquired LC-UV data is also given. (**i**): LC-UV chromatograms measured at 254 nm; (**ii**): LC-MS total ion chromatograms (TICs); (**iii**): extracted-ion chromatograms (XICs) from the LC-MS data of *m*/*z* values corresponding to some of the bioactive peaks in terms of matching peak shape and retention time (accurate masses calculated from the XICs are also given in the figures; matching XICs could be found only in some cases and are therefore not correlated with all PSCs); (**iv**): bioactivity chromatograms from the fluorescence enhancement-based AChBP tracer ligand displacement bioassay; (**v**): protein score chromatograms (PSCs) generated from the UniProt proteomic database. Toxin IDs are indicated next to their relevant PSC trace matching to a bioactive peak.

## Data Availability

The data presented in this study are available in this article and in the Appendix A.

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
