# Peer review of "Acetylcholine-Binding Protein Affinity Profiling of Neurotoxins in Snake Venoms with Parallel Toxin Identification"

_ijms, 2023, doi:10.3390/ijms242316769_

Round 1
Reviewer 1 Report
Comments and Suggestions for Authors
This manuscript introduces an integrated platform that combines HPLC, MS, HT venomics, and high content bioassay, by which snake venom 3FXs can be effectively profiled. The manuscript is interesting, and confronts the scope of the journal. However, some issues should be clarified before the consideration of publication.
1. The authors used TFA in the RPLC nanofractionation, but used FA in nanoLC, it’s better to be clarified. Generally, TFA has lower pH value, which is possibly able to change the conformation of inactivate the venom proteins. However, the FA may result in low separation resolution.
2. The manuscript indicates competition displacement of DAHBA from AChBP by fractionated toxins based on the decrease of RFU, however, the author didn’t define what’s the extent of decreasing can be considered as AChBP-toxin binding, such as in fig 2B, the negative peaks around at 23min and 40min were obvious, but they were not marked. What’s more, the sensitivity of the RFU bioassay was obscure, the weak peaks were possibly resulted from low concentration of toxins in the fractions.
3. In the section of results, the manuscript looks tedious because it comprise experimental methods, results and discussion, I suggest the authors to simplify this section.
4. In the abstract and introduction section, the authors said they want to investigate the therapeutic potential of AChBP as an alternative for treating snakebite. However, the whole manuscript actually aims at screening the long-chain 3Fxs from snake venom, the authors need to consider their phraseology in this respect.
5. Abstract: “The integrated method was established and applied to profiling venoms of three Elapid snakes (Naja mossambica, Ophiophagus hannah, Dendroaspis polylepis, Naja kaouthia, Naja haje and Bungarus multicinctus).” Three?
Line 501: A fractionation spotting time of 12 seconds for each fraction was used. For 384 well plate, 12 s or 6 s?
Line 156, and Ophiophagus hannah?
Comments on the Quality of English Languagenone
Reviewer 2 Report
Comments and Suggestions for Authors
In this study, the authors presented and applyed an analytical method for investigating the therapeutic potential of the acetylcholine binding protein (AChBP), an efficient nAChR mimic that can capture 3FTxs, for alternative treatment of elapid snakebites.
Comments
This is an interesting study. The manuscript is well-writing. The reviewer has only some minor concerns as follows:
1. How about the specificity of AChBP for nAChRs? It can be emphasized in the text. If the specificity is very high, in the Title, the “nicotinic acetylcholine receptors” can be emphasized for AChBP.
2. The presented reference numbes in the text can be revised, such as “[1]–[4]" changes to "[1-4]”, “[15], [16], [19], [20]" changes to "[15-20]”.
3. The limitations for this study can be described in the end of Discussion section.
